# A Novel Steganography Method for Infrared Image Based on Smooth Wavelet Transform and Convolutional Neural Network

**DOI:** 10.3390/s23125360

**Published:** 2023-06-06

**Authors:** Yu Bai, Li Li, Jianfeng Lu, Shanqing Zhang, Ning Chu

**Affiliations:** 1School of Computer Science and Technology, Hangzhou Dianzi University, Hangzhou 310018, China; baiyu@hdu.edu.cn (Y.B.); lili2008@hdu.edu.cn (L.L.); jflu@hdu.edu.cn (J.L.); 2Zhe-Jiang Shangfeng Special Blower Company Ltd., Shaoxing 312352, China; chuning1983@sina.com

**Keywords:** infrared images, steganography, convolutional neural network, CNN-based predictor, SWT, SRCNN, CNNP

## Abstract

Infrared images have been widely used in many research areas, such as target detection and scene monitoring. Therefore, the copyright protection of infrared images is very important. In order to accomplish the goal of image-copyright protection, a large number of image-steganography algorithms have been studied in the last two decades. Most of the existing image-steganography algorithms hide information based on the prediction error of pixels. Consequently, reducing the prediction error of pixels is very important for steganography algorithms. In this paper, we propose a novel framework SSCNNP: a Convolutional Neural-Network Predictor (CNNP) based on Smooth-Wavelet Transform (SWT) and Squeeze-Excitation (SE) attention for infrared image prediction, which combines Convolutional Neural Network (CNN) with SWT. Firstly, the Super-Resolution Convolutional Neural Network (SRCNN) and SWT are used for preprocessing half of the input infrared image. Then, CNNP is applied to predict the other half of the infrared image. To improve the prediction accuracy of CNNP, an attention mechanism is added to the proposed model. The experimental results demonstrate that the proposed algorithm reduces the prediction error of the pixels due to full utilization of the features around the pixel in both the spatial and the frequency domain. Moreover, the proposed model does not require either expensive equipment or a large amount of storage space during the training process. Experimental results show that the proposed algorithm had good performances in terms of imperceptibility and watermarking capacity compared with advanced steganography algorithms. The proposed algorithm improved the PSNR by 0.17 on average with the same watermark capacity.

## 1. Introduction

Recently, infrared images have been widely used in many real-time applications, such as pedestrian segmentation [1], salient object detection [2], object fusion tracking [3], and invisible clothing [4]. Compared with visible images, infrared images are also able to obtain temperature information of scenes and human bodies. Furthermore, they provide good visual effects in dark environments. Consequently, infrared images can provide new ideas for visible scenes in many different research fields.

Therefore, many researchers have proposed several image-processing algorithms based on infrared images. Most of these algorithms were based on fusing infrared images with visible images. The infrared images usually have the disadvantages of low texture details and more noise, which are the advantages of visible images. Thus, the idea of fusing both visible and infrared images is required to reconstruct a synthesized image containing prominent targets and abundant texture details. Many researchers have used multi-scale transforms to extract and match the features for both images. For example, Chen et al. [5] applied Laplace’s pyramid for infrared and visible images to obtain high- and low-frequency information, respectively. They applied regularization parameters and hence adjusted the scale of the features to achieve the flexible fusion of the two images. Yang et al. [6] used a sparse representation based on multi-scale decomposition to fuse out a base layer. This layer improved the accuracy of image integration. Unlike the multi-scale transform that was applied in [5,6], Li et al. [7] decomposed infrared images with visible images based on Latent Low-Rank Representation (LatLRR). The above-mentioned algorithms used traditional methods for infrared- and visible-images fusion. After Luo et al. [8] obtained infrared images using a Fourier Transform Infrared Spectrometer, they analyzed the changes in image details to grasp the evolution of specific microstructures.

Image-feature extraction has a very important position in all of the above algorithms. Thus, Wang et al. [9] improved the accuracy of feature extraction based on the ensemble-learning algorithm. Convolution Neural Networks (CNNs) are able to extract features of images very intelligently. Compared with traditional algorithms, CNN has better results in image-feature extraction. For example, Ao et al. [10] used CNN to extract images of patient-specific organs for local treatment. Fu et al. [11] proposed the depth-estimation algorithm for extracting simple and complex textures based on CNN. Wang et al. [12] investigated a multispecies transferable algorithm to improve the prediction accuracy of the CNN model. Zhang et al. [13] proposed a new classifier applied to an electroencephalogram for the overfitting phenomenon of CNN-model features. Deng et al. [14] applied CNNs to the field of natural-language processing to solve the challenge of non-action question and answer. Liu et al. [15] found that the global perception capability of CNNs needs to be improved. Therefore, they combined a Fast Fourier Transform with CNN to improve the performance of the model. Moreover, CNN has derived more novel models in various research areas in the last decade.

Furthermore, with the rapid development of machine learning and CNN, many neural-network-based algorithms have been applied for fusing infrared images and visible images. The Generative Adversarial Network (GAN) is widely used for infrared- and visible-image fusion [16,17,18]. Ma et al. [16] proposed the FusionGAN model, which is capable of fusing images with different resolutions. The FusionGAN generator first classified the features of infrared images and then fused them with visible images. The discriminator ensured that the fused images had more details of visible images. Additionally, a dual discriminator was added to the GAN for multi-resolution image fusion and was successfully applied for multi-modal medical images [17]. Ma et al. [18] presented the Generative Adversarial Network with Multiclassification Constraints (GANMcC)-based fusion model to estimate the distribution functions for both infrared and visible images. They introduced a specific content-loss function to constrain the generator. In addition to the classical GAN models, new customized models for infrared-image fusion have been presented [19,20]. Zhang et al. [19] proposed a low-time-complexity image-fusion CNN based on the Proportional Maintenance of Gradient and Intensity (PMGI) model. Their proposed model is capable of being used for different image-fusion tasks such as fusing visible and infrared images, fusing medical images, fusing multi-exposure images, and fusing multi-focus images [19]. Most of the existing deep-learning-based algorithms use convolution operation to extract local features but have not combined the network’s multi-scale characteristics and global dependencies, which may result in a disadvantage of target regions and texture details for fusing two types of images. Consequently, Wang et al. [20] combined the multi-scale transform with a self-encoder and applied an attention mechanism to fuse infrared images more accurately with visible images. They employed dense skip connections in both watermark embedding and watermark extraction to reuse the intermediate features of several layers and scales for fusing images [20].

The copyright protection of infrared images has captured many researchers’ attention recently. Image steganography is one of the commonly used algorithms to protect the copyright of images. In the literature, many image-steganography algorithms have been proposed to hide information based on pixel-prediction error. Li et al. [21] presented a reversible data-hiding algorithm that combined both Pixel-Value Ordering (PVO) and Prediction-Error Expansion (PEE). The PVO algorithm performs pixel prediction based on ordering pixels according to their values in an image block. However, their method failed to provide sufficient embedding positions by applying only one bin as the inner region. Ou et al. [22] extended the PVO method [21] and proposed PVO-k, where the maximum-value and minimum-value pixels in a block are taken as a unit to hide the watermark and were modified together to keep the PVO invariant. These PVO-based methods [21,22] performed splendidly, especially for low embedding capacity. However, a smaller block size should be applied to obtain a high embedding capacity, which causes more degradations in the quality of the watermarked image. Therefore, Wang et al. [23] improved the performance of the PVO-based methods and proposed a reversible data-hiding method based on dividing the cover image into a dynamic size of image blocks instead of applying same-sized blocks. Weng et al. [24] discovered that most of the improved PVO-based algorithms are applied to predict the pixels that occupy only a small fraction of the image block. Thus, they proposed a Dynamic Improved PVO (DIPVO) algorithm based on graded local complexity to dynamically hide information. Zhang et al. [25] proposed the Location-based PVO (LPVO) algorithm to solve the problem of irregular prediction-error distribution of PVO-based algorithms. The LPVO algorithm simultaneously considers the sequential location of pixel values, which makes more pixels available for image steganography.

In this paper, the infrared image is firstly divided into two sets: Dot sets and Cross sets. Secondly, we use the method of Super-Resolution based on deep CNN (SRCNN) [26] to pre-process half of the original image. This step refers to the image-super-resolution algorithm based on signal preprocessing proposed by Huang et al. [27]. Then, we apply Smooth Wavelet Transform (SWT) to extract high- and low-frequency sub-bands of the image. Thirdly, the CNN-based Predictor (CNNP) [28] is employed with a few attention-network layers to improve the accuracy of the pixel prediction. Finally, to make the process of the algorithm reversible, we proposed to train models that can recover the image and extract information simultaneously. The main contributions of this paper are as follows:Pixels in the same position have better prediction accuracy. Therefore, we employ SRCNN for pre-processing the Cross set of the infrared image to predict the Dot set of the original image, which reduces pixel-prediction errors.We apply SWT for extracting image-frequency-domain features to improve the prediction performance of the spatial CNN models. Furthermore, we employ the improved CNNP model on the image and frequency-domain features. The features in the spatial and frequency domains improve the accuracy of the training stage.An attention mechanism is added to the CNNP model. This allows the extracted features to be ranked according to their importance. The attention mechanism will also speed up the model training and improve the performance of the testing stage of the model.

The rest of this paper is organized as follows: Section 2 summarizes the related works, including the basic idea behind the SRCNN model and the CNNP model. Section 3 presents the proposed method in detail. The experimental results and performance analysis of the proposed algorithm compared with other algorithms are shown in Section 4. Finally, the conclusion and some future directions can be found in Section 5.

## 2. Related Works

Recently, CNN-based image-steganography algorithms have become a popular research direction. Luo et al. [29] used DenseNet to extract high-level semantic features of each block instead of low-level features and then hide information by using DCT. Sharma et al. [30] combined steganography with cryptography, and they used CNN for image steganography and image recovery. Similarly, Hu et al. [28] proposed a CNN-based Predictor (CNNP). They applied the same structure of the CNN model for image steganography and image recovery, but the training set was different. Hu et al. [31] improved the structure of the model and introduced the concept of chunking to improve the accuracy of pixel prediction. The CNN-based models presented by Hu et al. [28] and Hu et al. [31] were very small and the training process could be run on the CPU. Furthermore, Panchikkil et al. [32] applied Arnold-transform-based data hiding to improve the performance of image recovery, where each block underwent some complex scrambling processes based on the watermark information required for embedding in the selected block. Their method has two segments: a watermark-embedding segment that hides confidential information in the cover image, and a watermark-extraction segment that recovers the original cover image and extracts the watermark information [32].

Overall, CNN-based algorithms have excellent performance in image steganography in the spatial domain, but the combination with the traditional frequency-domain transforms should not be neglected. The algorithms based on the spatial domain directly process the pixels in the image, and the frequency domain is a variety of high- or low-frequency sub-bands that are created by an image-transform algorithm. With this in mind this, Liu et al. [33] combined U-Net with Haar Wavelet Transform (HWT). HWT was applied to extract high- and low-frequency features of the image. Then, the U-Net was used to hide information in the original image. In this paper, the model of Super-Resolution based on deep CNN (SRCNN) [26] was applied to pre-process the infrared image and combine it with its corresponding SWT sub-bands. Furthermore, we applied the CNNP model [28] on the image and frequency-domain features. The features in the spatial and frequency domains improved the accuracy of the training step and hence improved the prediction accuracy of the proposed algorithm.

### 2.1. SRCNN Model

Dong et al. [26] presented the first machine-learning framework of SRCNN for single-image Super Resolution (SR). In the pre-processing process, they applied bilateral interpolation for upsampling the original image $I_{co}$ to the desired size. Bilinear interpolation enables fast and relatively accurate upsampling of images. The image at this Dot set is defined as the interpolated image $I_{bi}$. The general structure of the SRCNN model is shown in Figure 1.

SRCNN aims to recover an interpolated image $I_{bi}$ from an image $F\left(I_{bi}\right)$ that is mostly similar to a high-definition (HD) image $I_{sr}$. The algorithm should ensure that the output result of $F\left(I_{bi}\right)$: HD image $I_{sr}$ as close as possible to the interpolated image $I_{bi}$. The SRCNN algorithm can be divided into three main steps: patch extraction, nonlinear mapping, and image reconstruction. Intensive patch extraction is a very popular strategy for image recovery. Moreover, feature extraction and representation of the patches in a reasonable way plays a key role in the image-SR algorithm. Thus, the first layer of the SRCNN model is represented in Equation (1) as follows:(1)$$F_{1}\left(I_{bi}\right)=\max\left(0,W_{1}*I_{bi}+B_{1}\right)$$
where $W_{1}$ is a filter for extracting features, $B_{1}$ is a parameter for adjusting the bias, and ∗ represents the convolution operation. First, $W_{1}$ extracts the features from the interpolated image $I_{bi}$. Then, the ReLU activation function is applied to correct the extracted features. Hence, plenty of features can be obtained in the patch-extraction step, whereas in the nonlinear-mapping step, more features are further extracted and then classified. The calculation process is shown in Equation (2) as follows:(2)$$F_{2}\left(I_{bi}\right)=\max\left(0,W_{2}*F_{1}\left(I_{bi}\right)+B_{2}\right)$$
where $W_{2}$ is a filter for extracting features, and $B_{2}$ is a parameter for adjusting the bias. Theoretically, more convolutional layers can be added in this step to improve the performance of the model, but the complexity of the model and the training cost will also be increased. In transform-based algorithms of the frequency domain, the predicted overlapping blocks of SR pixels are usually averaged to generate a good image recovery. In order to improve the image-prediction accuracy of the model, we need to increase the number of convolutional layers to improve the perceptual field of the convolutional kernel. This reduces the error of the model-output results and is also more beneficial for the subsequent information-hiding step. Inspired by traditional algorithms, the SRCNN model incorporates a dialogue layer. Equation (3) shows how this dialogue layer is computed.
(3)$$F\left(I_{bi}\right)=W_{3}*F_{2}\left(I_{bi}\right)+B_{3}$$

Let the parameters to be learned by the SRCNN model be: $\Theta =\left\{W_{1},W_{2},W_{3},B_{1},B_{2},B_{3}\right\}$. Then, for the set of HD images $I_{sr}$ and the corresponding interpolated images $I_{bi}$, the loss function is defined as the Mean-Square-Error (MSE) function as follows:(4)$$L_{SR}=\mathrm{Average}\sum {\left\|F\left(I_{bi},\Theta \right)-I_{sr}\right\|}^{2}$$

The MSE loss function tends to result in high-quality images for the algorithm. The SRCNN model is obtained when the value of the loss function meets the requirements of the model training.

### 2.2. CNNP Model

The CNNP model [28] is a CNN-based framework for image steganography. The structure of the model is very simple and does not require the GPU server during the model-training phase. Furthermore, it makes full use of the excellent extraction capability of CNN for global features, which makes the watermark capacity exceed many traditional image-steganography algorithms. The CNNP model firstly divides the images into two groups equally, as shown in Figure 2, which are named the Cross collection and the Dot collection. The algorithm then predicts the Dot collection $I_{D}$ by the Cross collection $I_{C}$. Next, the predicted Dot collection ${I^{\prime }}_{D}$ is input into the CNNP model to predict the Cross collection $I_{C}$. Finally, they combine the predicted Cross collection ${I^{\prime }}_{C}$ with the predicted Dot collection ${I^{\prime }}_{D}$ to obtain the full predicted image. The main structure of the CNNP model is shown in Figure 3.

The model consists of several convolutional blocks. Each convolutional block consists of K×K convolutional layers, a LeakyReLU activation function, and 3 × 3 convolutional layers. The algorithm applies a two-stage information-hiding approach. Firstly, the error between the Dot collection $I_{D}$ and predicted Dot collection ${I^{\prime }}_{D}$ needs to be calculated, and then the appropriate prediction error (usually 0) is selected for image steganography. The information-hiding process for the Cross-set images $I_{C}$ is the same as that for the Dot collection $I_{D}$. Furthermore, the algorithm flow of CNNP is reversible. In the flow of image recovery, the CNNP model is applied first to recover the Cross collection $I_{C}$, and then the Cross collection $I_{C}$ is input into the model to recover the Dot collection $I_{D}$.

## 3. Proposed Algorithm

The proposed method contains three main aspects: image steganography, image recovery, and information extraction. For the information-hiding process, we firstly divide the input infrared image into two sets: the Cross collection $I_{C}$ and the Dot collection $I_{D}$, as shown in Figure 2. Then, the watermarked Dot collection $I_{D,W}$ is used to predict the Cross collection $I_{C}$. The prediction process is performed by SRCNN and SWT for the Cross collection $I_{C}$ and Dot collection $I_{D}$, respectively. The CNNP model for predicting infrared images is also incorporated into the attention layers to improve the accuracy of the prediction. After the full predicted image $I_{W}$ is obtained, the watermarking information is embedded into the infrared image. Furthermore, information-extraction and image-recovery processes are then performed simultaneously. The watermarked Cross collection $I_{C,W}$ is first applied to predict the Dot collection $I_{D}$ and the watermark is extracted based on the prediction error. The process for recovering the Cross collection $I_{C}$ and extracting the information is done in the same way.

### 3.1. Image Steganography

In image-steganography step, two CNN-based models are employed: the SRCNN model and the CNNP model. Moreover, SWT is employed to extract the frequency-domain features of the infrared image. The overall structure of the proposed steganography algorithm is shown in Algorithm 1 and Figure 4a. Figure 4b indicates the process of SSCNNP (SRCNN followed by CNNP based on the SE layer).
**Algorithm 1: General flow of the algorithm**1: Image is divided into collections of Cross/Dot IC,co/ID,co.2: SRCNN preprocesses the IC,co/ID,co.3: SWT decomposes the result of SRCNN ID,sr/IC,sr.4: SE-CNNP predicts the ID,co/IC,co.5: Reversible hiding or extraction of information.6: Steps 2~5 for the other half of the image.

Image steganography can be divided into four main steps: initial image prediction, SWT decomposition, CNNP-based image prediction, and difference-histogram translation. These segments are carefully described in the following subsections.

#### 3.1.1. Initial Image Prediction

Inspired by the SRCNN model for image SR, the initial prediction stage of the image in the proposed method aims to take up as few resources as possible to predict the information of neighboring pixels accurately. Empirically, the CNN model is trained better for input and output images with the same pixel positions. Therefore, if we apply the Cross/Dot $I_{C}$/$I_{D}$ collections to predict each other, the prediction error can be reduced. Therefore, the original collections of Cross/Dot $I_{C,co}$/$I_{D,co}$ are required to be accurately pre-processed before predicting the images. Since the SRCNN model is applied for pre-processing, the Dot/Cross-set images here are denoted by $I_{D,sr}$/$I_{C,sr}$, respectively. As shown in Figure 1, the SRCNN model receives the original Cross/Dot-set image $I_{C,co}$/$I_{D,co}$ and outputs the SR collections of Dot/Cross $I_{D,sr}$/$I_{C,sr}$, respectively. Generally, a 9 × 9 convolutional layer first transforms the original Cross/Dot images $I_{C,co}$/$I_{D,co}$ into 64 feature maps. Then, the model can be applied to more nonlinear scenes by adding nonlinear factors using the ReLu activation function. Finally, two 5 × 5 convolutional layers are applied to extract the features from the image and recover the image as a single-channel infrared image. Moreover, the SRCNN model updates the learning parameters $\theta_{sr}$ to reduce the error between the SR Dot/Cross images ($I_{D,sr}$/$I_{C,sr}$) and the original Dot/Cross images ($I_{D,co}$/$I_{C,co}$). Finally, the classical MSE function is applied by the loss function $L_{sr}$ as follows:(5)$$L_{sr}=\left\{ \begin{array}{l} \mathrm{MSE}\left(I_{D,co},I_{D,sr}\right)=\mathrm{MSE}\left(I_{D,co},\mathrm{SR}\left(\theta_{sr},I_{C,co}\right)\right)\\ \mathrm{MSE}\left(I_{C,co},I_{C,sr}\right)=\mathrm{MSE}\left(I_{C,co},\mathrm{SR}\left(\theta_{sr},I_{D,co}\right)\right) \end{array}\right.$$

#### 3.1.2. SWT Decomposition

CNN model-based algorithms usually extract features from images in the spatial domain. In this paper, we propose to combine an infrared image with its four SWT low-/high-frequency sub-bands inside a tensor and then feed that tensor into the CNNP model for image prediction. In contrast to other wavelet transforms, SWT is a non-downsampling transform, i.e., all sub-bands are equal in size to the original image [34]. Low-pass and high-pass SWT filters keep all low- and high-frequency components the same size by padding with 0 coefficients in the process of coefficient processing. Taking the SR Dot-set image $I_{D,sr}$ as an example, the SWT decomposition can be shown as in the following Equations (6)–(9):(6)$$\mathrm{LL}\left(i,j\right)=\sum_{k,m}h_{k}h_{m}I_{D,sr}\left(i+k,j+m\right)$$
(7)$$\mathrm{HL}\left(i,j\right)=\sum_{k,m}g_{k}h_{m}I_{D,sr}\left(i+k,j+m\right)$$
(8)$$\mathrm{LH}\left(i,j\right)=\sum_{k,m}h_{k}g_{m}I_{D,sr}\left(i+k,j+m\right)$$
(9)$$\mathrm{HH}\left(i,j\right)=\sum_{k,m}g_{k}g_{m}I_{D,sr}\left(i+k,j+m\right)$$
where LL is a low-frequency sub-band, and HL, LH, and HH are high-frequency sub-bands generated by SWT decomposition. g and h are the filters of the SWT transform for different sub-bands. In this paper, the images in the spatial domain and low-/high-frequency features in the frequency domain were jointly used as inputs to the CNNP model to improve the convergence speed and prediction accuracy of the model.

#### 3.1.3. CNNP-Based Image Prediction

In this paper, we employed a Squeeze-and-Excitation Network (SENet) [35] to improve the accuracy of the CNNP model. Compared with many CNN-based models that modify the spatial dimension to improve the image-prediction accuracy, SENet instead increases the performance by modeling the relationship of feature channels. Usually, an SE layer takes up few resources and largely improves the convergence speed and performance of the model. In this paper, we modified the CNNP model by applying the SE layer to improve the prediction accuracy of the CNNP model. The structure of the modified CNNP model is shown in Figure 5.

The input tensor consists of an infrared image and four frequency components of SWT (LL, HL, LH, LL). The SWT sub-bands need to be modified according to the type of infrared image. For example, if the infrared image is the SR collection of Dot $I_{D,sr}$, the Cross-set position of all SWT sub-bands should be set to 0. Consequently, the coefficient of the Dot-set position has to be changed to 0 if the infrared image is the SR collection of Cross $I_{C,sr}$. We first extracted the features of the input tensor by applying 3 × 3, 5 × 5, and 7 × 7 convolutional blocks. Each convolution block consisted of two K × K convolution layers and a LeakyReLU activation function, as shown in Figure 5b. The LeakyReLU function is an improved version of the classical ReLu activation function, and it has a small slope output for negative inputs. Since the derivative is essentially non-zero, the number of silent neurons is reduced substantially, which allows for gradient-based learning. The LeakyReLU function can solve the problem in which the neurons do not learn if the ReLu function has negative values. However, the training time is increased dramatically. Therefore, we summed up the extracted features and recorded the importance of each feature layer by the SE layer. The SE layer quantifies the importance of the features and visualizes it on the features. Finally, we predicted the final infrared image by using two 3 × 3 convolutional blocks to prevent the CNNP model from gradient disappearance or explosion. One of the convolution blocks was referenced in the form of a residual block, which was connected to the SE layer. By using the SR Dot/Cross ($I_{D,sr}$/$I_{C,sr}$) collections as inputs to the model for training, the loss function $L_{CP}$ could be defined as in Equation (10) as follows:(10)$$L_{CP}=\left\{ \begin{array}{l} \mathrm{MSE}\left(I_{D,co},I_{D,pre}\right)=\mathrm{MSE}\left(I_{D,co},\mathrm{CNNP}\left(\theta_{cp},I_{D,sr}\right)\right)\\ \mathrm{MSE}\left(I_{C,co},I_{C,pre}\right)=\mathrm{MSE}\left(I_{C,co},\mathrm{CNNP}\left(\theta_{cp},I_{C,sr}\right)\right) \end{array}\right.$$
where $\theta_{cp}$ is the parameter of the CNNP model. Similar to Section 3.1.1, the CNNP model also uses the MSE function as a loss function. When the loss function $L_{CP}$ is at a very low value for a long time, it means that the parameters of the model have reached the best case.

#### 3.1.4. Histogram Shifting

Each digital image has its own histogram, including infrared images. The histogram visualizes the number of pixels in the image for each pixel value. Due to its small computational cost and its translation, scaling, and rotation invariance, the image histogram is widely applied in many fields of digital-image processing. In image steganography, histogram shifting is one of the classical reversible information-hiding algorithms [36,37,38]. In this paper, we propose using the histogram-shifting algorithm for reversible image steganography based on the CNNP model to predict the complete final image. Different from traditional histogram-shifting-based algorithms, we selected the prediction error of infrared images to count the histogram. Figure 6a shows a histogram of the prediction error for an infrared image. This image was randomly selected from the test set.

From Figure 6a, we can see that the proposed SSCNNP model had a good prediction accuracy, where the peaks of the prediction-error histogram were very high at the position 0. This means that we do not need to remember the peak points of the histogram, as in the traditional histogram-shifting algorithm. The peak point of the histogram must be located at level 0 because the prediction error of the algorithm concentrates at 0. The information-hiding algorithm is slightly different depending on the location of the nearest zero point to the peak. For example, in Figure 6a, the zero value on the right side of the histogram was closest to the peak point (the pixel value between the two points was the lowest), so the equation for the proposed image steganography is shown in Equation (11) as follows:(11)$$P_{w}\left(i\right)=\left\{ \begin{array}{l} P_{co}\left(i\right),\;\;\;\;\;\;\;\;\;\mathrm{if}\;P_{co}\left(i\right)=0,\;w\left(j\right)=0\\ P_{co}\left(i\right)+1,\;\;\;\;\;\mathrm{if}\;P_{co}\left(i\right)=0,\;w\left(j\right)=1\\ P_{co}\left(i\right)+1,\;\;\;\;\;\mathrm{if}\;P_{co}\left(i\right)>0 \end{array}\right.$$
where $P_{co}$ and $P_{w}$ are the original and watermarked prediction errors, respectively. The prediction error is the difference between the pixel predicted by the model and the original pixel. The pixel prediction error is inversely proportional to the capacity of information hiding. *i* is the index of the pixel. w is the watermarking information to be hidden, and j is watermark’s index. Conversely, if the nearest zero point is located to the left of the peak, the proposed image steganography can be determined as in Equation (12) as follows:(12)$$P_{w}\left(i\right)=\left\{ \begin{array}{l} P_{co}\left(i\right),\;\;\;\;\;\;\;\;\;\mathrm{if}\;P_{co}\left(i\right)=0,\;w\left(j\right)=0\\ P_{co}\left(i\right)-1,\;\;\;\;\;\mathrm{if}\;P_{co}\left(i\right)=0,\;w\left(j\right)=1\\ P_{co}\left(i\right)-1,\;\;\;\;\;\mathrm{if}\;P_{co}\left(i\right)<0 \end{array}\right.$$

Briefly, the algorithm strives to minimize the number of pixels that are changed with an equal number of watermarks. Furthermore, the positions of these zeros remain unremembered. For example, in Figure 6b, since the nearest zero point was located to the right of the peak, the watermark-containing histogram had more pixels to the right of 0. Therefore, the direction of the histogram translation could be selected as the right side.

### 3.2. Image Recovery and Information Extraction

Image recovery and information extraction are performed simultaneously. Furthermore, the information-extraction process is similar to the steps of hiding information. The SRCNN initially predicts the infrared image. Moreover, low-/high-frequency sub-bands of SWT decomposition are fed into the improved CNNP model along with the infrared image itself. For information extraction, the proposed method first uses the watermarked collection of Dot $I_{D,W}$ to predict the collection of Cross $I_{C}$. The result obtained from the prediction and the watermarked collection of Cross $I_{C,W}$ calculates the prediction error to extract the information. While the information is extracted, the collection of Cross $I_{C}$ is also recovered. The flow structure of the proposed image recovery and information extraction is shown in Figure 7.

#### 3.2.1. SRCNN and SWT Pre-Processing

Firstly, SRCNN is applied to pre-process the watermarked collection of Dot/Cross ($I_{D,W}$/$I_{C,W}$). The output of the model is recorded as the watermarked collection of SR Cross/Dot ($I_{C,srw}$/$I_{D,srw}$, respectively). Secondly, SWT decomposes the watermarked collections of SR Cross/Dot $I_{C,srw}$/$I_{D,srw}$, and hence one low-frequency sub-band (LL) and three high-frequency sub-bands (HL, LH, and HH) are obtained. The SWT sub-bands together with the watermarked collections of SR Cross/Dot $I_{C,srw}$/$I_{D,srw}$ are fed into the CNNP model to improve the convergence speed and prediction performance of the model.

#### 3.2.2. CNNP Recovery and Extraction

Firstly, the watermarked collections of SR Cross/Dot ($I_{C,srw}$/$I_{D,srw}$) are fed into the improved CNNP model. The addition of the SE layer in the embedding stage allows the CNNP model to obtain a more accurate predicted collection of Cross/Dot ($I_{C,pre}$/$I_{D,pre}$). Secondly, the prediction-error difference between the predicted collections of Cross/Dot ($I_{C,pre}$/$I_{D,pre}$) and the watermarked collections of Cross/Dot ($I_{C,W}$/$I_{D,W}$) is calculated. Equation (13) and Equation (14) are the equations for image recovery and watermark extraction when the pixel-prediction error ≥ 0 and ≤ 0, respectively. Based on the difference-histogram translation mentioned in Section 3.1.4, the formulas for extracting information in both cases are defined as follows:(13)$$\left\{ \begin{array}{l} P_{co}\left(i\right)=P_{w}\left(i\right),\;\;\;\;\;w\left(j\right)=0,\;\;\;\;\;\mathrm{if}\;P_{w}\left(i\right)=0\\ P_{co}\left(i\right)=P_{w}\left(i\right)-1,\;w\left(j\right)=1,\;\;\;\;\;\mathrm{if}\;P_{w}\left(i\right)=1\\ P_{co}\left(i\right)=P_{w}\left(i\right)-1,\;\;\;\;\;\;\;\;\;\;\;\;\;\;\;\;\;\;\mathrm{if}\;P_{w}\left(i\right)>1 \end{array}\right.$$
(14)$$\left\{ \begin{array}{l} P_{co}\left(i\right)=P_{w}\left(i\right),\;\;\;\;\;w\left(j\right)=0,\;\;\;\;\;\mathrm{if}\;P_{w}\left(i\right)=0\\ P_{co}\left(i\right)=P_{w}\left(i\right)+1,\;w\left(j\right)=1,\;\;\;\;\;\mathrm{if}\;P_{w}\left(i\right)=-1\\ P_{co}\left(i\right)=P_{w}\left(i\right)+1,\;\;\;\;\;\;\;\;\;\;\;\;\;\;\;\;\;\;\mathrm{if}\;P_{w}\left(i\right)<-1 \end{array}\right.$$

Next, the same process is applied to predict the collection of Dot $I_{D}$ by using the watermarked collection of Cross $I_{C,W}$. While the watermark information is extracted from the watermarked collection of Dot $I_{D,W}$, the collection of Cross $I_{C}$ is also recovered. Finally, the algorithm can obtain both the extracted watermark information w and the original infrared image I.

## 4. Experimental Results

### 4.1. Experimental Configuration

We conducted our experiments on 113 infrared images from wind turbines from Zhejiang Shangfeng Group Co., Ltd., Shaoxing, Zhejiang, China. The size of each infrared image was 256 × 256. The dataset was partitioned into 90 images in the training set and 23 images in the testing set. They were all infrared images captured from all angles in normal or faulty conditions. The shape of the wind turbine differed greatly from angle to angle, resulting in some variation in the images of the dataset. This effectively alleviated the problem of model overfitting. If researchers would like to obtain this dataset, please contact Zhejiang Shangfeng Group Co., Ltd. Additionally, to demonstrate the generality of the proposed algorithm on infrared images, a substation-power-equipment dataset was purchased online and used for the experiments. The substation-power-equipment dataset contained 100 infrared images, from which 80 infrared images were selected as a training set and 20 infrared images as a testing set. Figure 8 shows four infrared images from the substation-power-equipment dataset. The proposed algorithm was trained on Dell OptiPlex 7070, with an Intel(R) Core(TM) i7-9700 CPU. The SRCNN and SSCNNP models used are very simple, so the CPU was sufficient to train and test the models. The codes were implemented in Matlab 2019a and Pycharm 2020.1.2.

### 4.2. Prediction Accuracy

To evaluate the prediction-accuracy performance of the proposed algorithm, we used the mean absolute value, variance, and MSE of the image-prediction errors for the proposed SSCNNP model, the CNNP model, and three state-of-the-art algorithms: BIP [39], MEDP [40], and GAP [41]. Table 1 shows that the proposed SSCNNP had a good prediction accuracy compared with CNNP, BIP [39], MEDP [40], and GAP [41]. For the three evaluation indicators shown in Table 1 below, the lower the values, the more accurate the prediction was.

As we can see from Table 1, the absolute value of the mean prediction error of the proposed SSCNNP model was only 2.04, which was lower than that of the CNNP model (2.72), BIP (4.19), MEDP (5.24), and GAP (6.59). Moreover, the variance of the proposed SSCNNP model was only 28.64, which was lower than that of CNNP model (37.15), BIP (62.48), MEDP (103.19), and GAP (136.42). For MSE, the proposed SSCNNP model achieved 36.91, which was lower than the CNNP model (59.37), BIP (93.60), MEDP (148.43), and GAP (207.32). Table 1 indicates that the proposed SSCNNP model outperformed the others on image-prediction accuracy.

Figure 9 shows the prediction errors of the proposed SSCNNP, CNNP, BIP [39], MEDP [40], and GAP [41] for the four infrared images shown in Figure 8. The higher the vertical coordinate of the model in the figure at the position where the prediction error was 0, then the better the model was. For better visualization, only the prediction errors within the range [–10, 10] are shown. From Figure 9, it was found that the prediction error histogram of the proposed SSCNNP model outperformed the other methods. In particular, the proposed SSCNNP model had more pixels with accurate predictions for the prediction error in the range [–1, 1]. The BIP, MEDP, and GAP algorithms use a single convolution-kernel feature extraction and pixel prediction. Furthermore, the CNNP model applies three convolution kernels, 3 × 3, 5 × 5, and 7 × 7, at the same time to extract the features more comprehensively. However, the proposed SSCNNP model applies SRCNN to unify the positions of the predicted carrier pixels with the pixels to be predicted. Moreover, SWT also facilitates the training of the model by extracting low- and high-frequency sub-bands of the image. Additionally, the SE layer improves the training accuracy of the CNNP model. Therefore, the experimental results shown in Figure 9 demonstrate that the proposed SSCNNP model had better prediction performance compared with the other algorithms. This provided a good basis for the subsequent reversible steganography.

### 4.3. Information-Hiding Performance

In this section, we introduce a performance analysis of the proposed SSCNNP compared with CNNP, LPVO [42], IPPVO [25], and BLTM [43] for information hiding with different watermark capacities on the two test datasets, as listed in Table 2 and Table 3.

The IPPVO algorithm adaptively selects the size of the convolution kernel according to the texture complexity of the current pixel block. The LPVO algorithm can quickly and accurately select multiple histograms for panning to hide information. The IPPVO and LPVO are commonly related to the proposed algorithm, so it can be applied for comparison. Figure 10a–d show the PSNR values of the four infrared images shown in Figure 8 with various watermarking capacities on the wind-turbine dataset. The higher the PSNR value, the closer the watermarked image $I_{w}$ was to the original image I, i.e., the better the imperceptibility of the image-steganography algorithm. The watermarking capacity goes from 2000 bits to 30,000 bits and increases by 2000 bits at a time.
(15)$$\mathrm{PSNR}=10\times \log_{10}\frac{M\times N\times 255^{2}}{\left[\sum_{i=1}^{M}\sum_{j=1}^{N}{\left(I-I_{w}\right)}^{2}\right]/\left(M\times N\right)}$$

In Equation (15), M and N are the length and width of the original image I, respectively, and in this paper M=N=256. The higher the PSNR value, the closer the watermarked image was to the original image I.

#### 4.3.1. Performance on Wind-Turbine Dataset

The experimental results shown in Figure 10 demonstrate that the watermarked images with the proposed algorithm had higher PSNR values than those of the wind-turbine dataset. Table 2 shows the PSNR values of the proposed algorithm compared with CCNP [28], LPVO [42], IPPVO [25], and BLTM baiyu@hdu.edu.cn 43] when the watermarking capacity was 10,000 bits and 20,000 bits. When the watermark capacity was 10,000 bits, the average PSNR value of the proposed SSCNNP-based image steganography algorithm was 59.27 dB, which was better than that of the BLTM (58.88 dB), IPPVO (58.91 dB), LPVO (58.93 dB), and CNNP (59.14 dB) based on image-steganography algorithms. Furthermore, when the watermarking capacity was increased to 20,000 bits, the average PSNR value for our algorithm was still higher. The better imperceptibility of the proposed SSCNNP model could be directly observed by averaging the PSNR values of four images.

#### 4.3.2. Performance on Substation-Power-Equipment Dataset

The substation-power-equipment dataset had many different types of infrared images. These different kinds of infrared images may have had different characteristics. Therefore, we demonstrated the performance of the proposed algorithm for the infrared images in the substation-power-equipment dataset through comparison experiments. The comparison experiments use the four infrared images of substation power equipment shown in Figure 8. The variation of the PSNR values of the algorithm with different watermarking capacities is shown in Figure 11. The experimental results demonstrate that the proposed SSCNNP model had the highest PSNR values for all watermarking capacities. This also proves that the proposed algorithm had good imperceptibility compared to the other state-of-the-art algorithms. Because the watermark capacity of individual images was less than 20,000, Table 3 shows the PSNR values of the four algorithms when the watermarking capacity was 5000 bits and 10,000 bits.

When the watermark capacity was 5000 bits, the average PSNR value of the proposed algorithm was 60.08 dB, which was higher than the BLTM (59.34 dB), IPPVO (59.34 dB), LPVO (59.52 dB), and CNNP (59.86 dB) image-stenography algorithms. The image-steganography algorithm based on the proposed SSCNNP model still had the highest average PSNR value when the watermarking capacity was 10,000 bits. Overall, the experimental results demonstrate that the proposed algorithm had better performance in terms of imperceptibility compared with the state-of-the-art algorithms.

## 5. Conclusions

In this paper, a novel steganography algorithm for infrared images is proposed based on the SRCNN and CNNP models. The pre-processing model is the SRCNN model, which is able to predict image pixels more accurately. Furthermore, the proposed model combines the frequency domain of SWT with the spatial domain of the CNNP model to reduce the prediction error. Experiments show that the proposed SSCNNP model had better prediction performance for both classical-prediction-based algorithms and novel-CNNP-based models. Therefore, the imperceptibility and watermark capacity were also improved for infrared images. Moreover, the SRCNN and CNNP models are both lightweight CNN models, and they can be combined with each other in a harmonious way. In future work, we will try to introduce more lightweight CNN models and wavelet transform in the frequency domain to reduce the requirements of hardware equipment and improve the performance of the algorithm at the same time. Moreover, the experimental images should be more abundant and closer to real-time images to ensure the practicality of the algorithm.

However, the infrared applied in this paper was only for the device, so in the future we hope to extend the study to a larger range of infrared images. In addition to image steganography, robust image-watermarking algorithms are also well worth exploring.

## Figures and Tables

**Figure 1 sensors-23-05360-f001:**
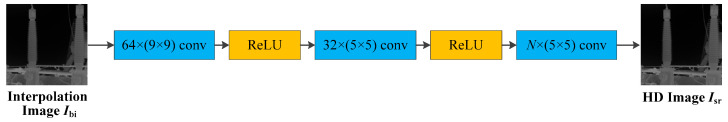
The general structure of the SRCNN model [18].

**Figure 2 sensors-23-05360-f002:**
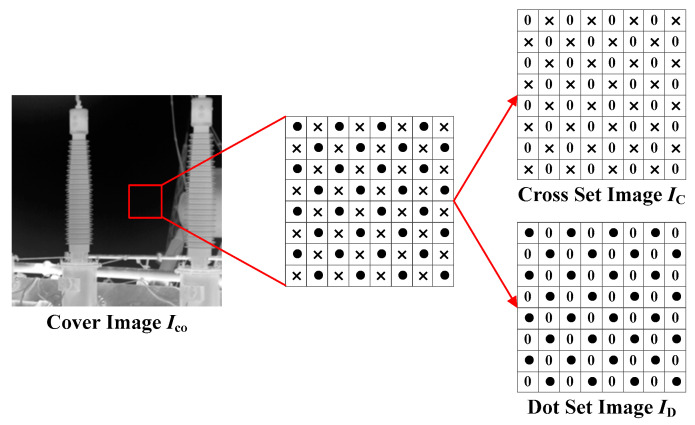
Division of Dot set images and Cross set images. 0 represents setting the pixel value to 0. Dot and Cross are used to distinguish pixels in two different positions.

**Figure 3 sensors-23-05360-f003:**
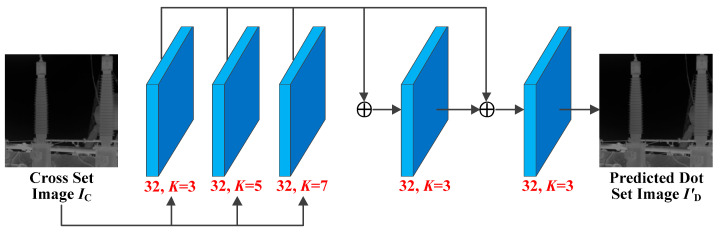
Structure of the CNNP model. “+” means adding the elements at the corresponding positions of two features.

**Figure 4 sensors-23-05360-f004:**
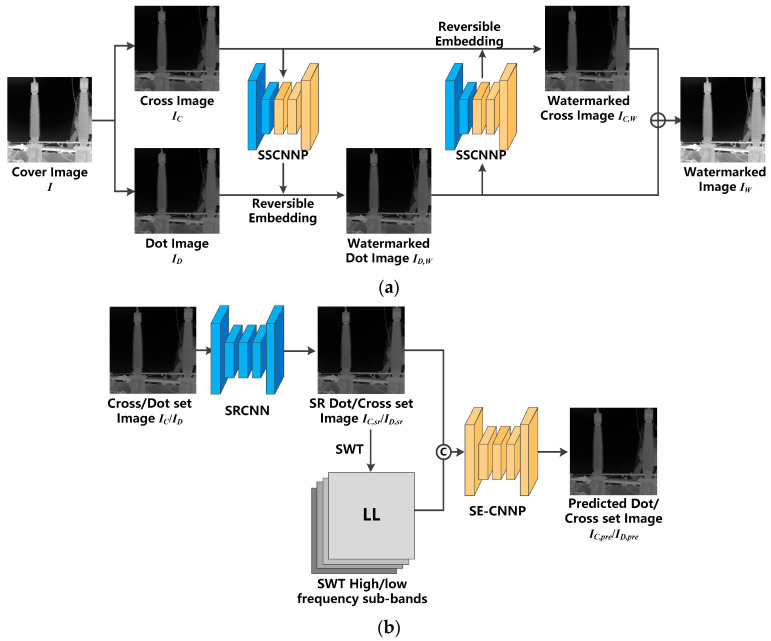
General flow diagram of the proposed image steganography. (**a**) The inputs of the SSCNNP model are the Cross image $I_{C}$ and the watermarked Dot image $I_{D,W}$. These two model outputs are computed with the error of the Dot image $I_{D}$ and the Cross image $I_{C}$. Finally, reversible embedding is performed based on the error. Two types of watermarked images form the watermarked image $I_{W}$. (**b**) Low-frequency sub-bands are signals with lower frequencies and generally preserve the contours of the image. The high-frequency signal is the signal with the high frequency and generally preserves the noise and details of the image. (**a**) The proposed image-steganography flow steps; (**b**) the structure of the proposed SSCNNP (SRCNN followed by CNNP) algorithm. ‘C’ means concatenating features according to the third dimension.

**Figure 5 sensors-23-05360-f005:**
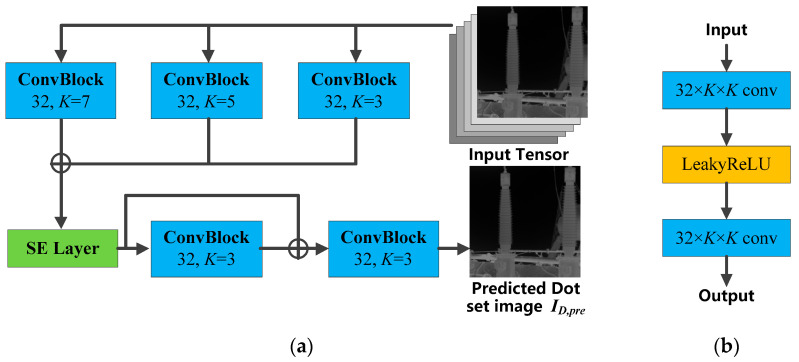
The modified CNNP model. (**a**) Structure diagram of the CNNP model with the SE layer, (**b**) structure diagram of the convolutional block of the CNNP model.

**Figure 6 sensors-23-05360-f006:**
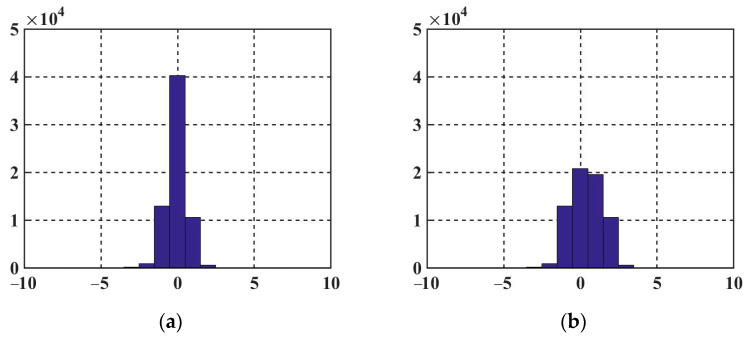
Prediction-error histogram. The coordinate of the horizontal axis is the image prediction error. The coordinate of the vertical axis is the number of pixels corresponding to the prediction error. (**a**) The more pixels in the original image that have a prediction error of 0, the better the performance of the algorithm. (**b**) The watermarked image assigns pixels with an original error of 0 to positions with prediction errors of 0 and 1. (**a**) Original infrared image; (**b**) watermarked infrared image.

**Figure 7 sensors-23-05360-f007:**
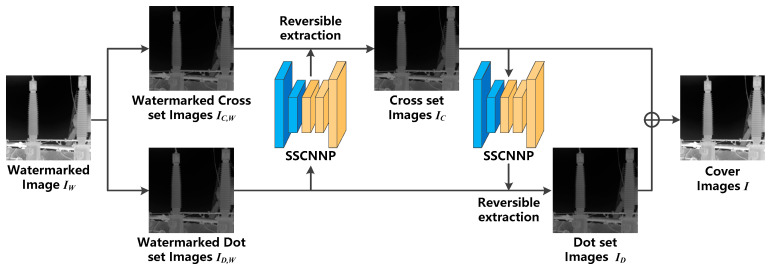
Flow diagram for image recovery and information extraction.

**Figure 8 sensors-23-05360-f008:**
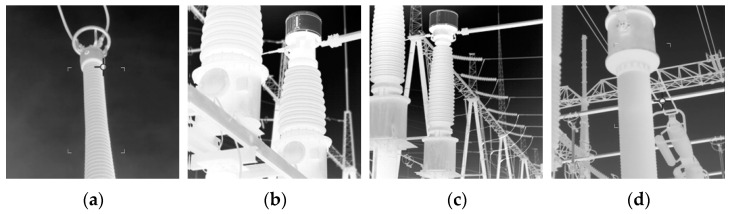
Four infrared images from the testing set of the substation-power-equipment dataset. (**a**–**d**) are four types of substation-power-equipment.

**Figure 9 sensors-23-05360-f009:**
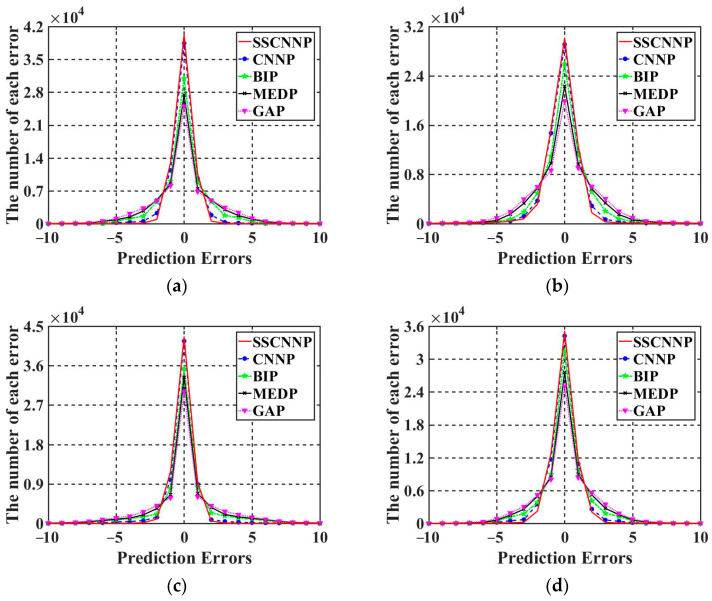
The prediction errors for the four infrared images shown in Figure 8. (**a**–**d**) correspond to the four subfigures in Figure 8, respectively.

**Figure 10 sensors-23-05360-f010:**
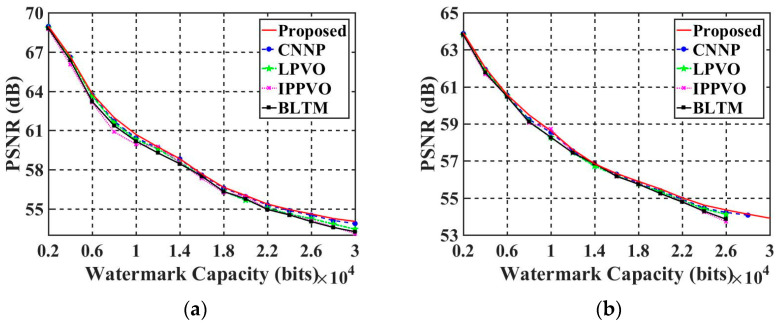
PSNR values for different watermarking capacities on the wind-turbine dataset. (**a**–**d**) corresponds to four wind-turbines from Zhejiang Shanghai Group Co., Ltd., Shaoxing, Zhejiang, China.

**Figure 11 sensors-23-05360-f011:**
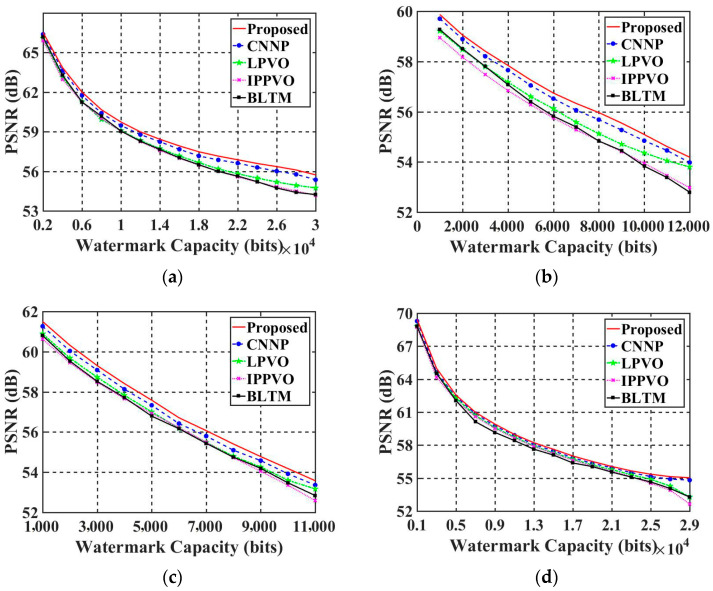
PSNR values for two watermarking capacities on the substation-power-equipment dataset. (**a**–**d**) correspond to the four subfigures in Figure 8, respectively.

**Table 1 sensors-23-05360-t001:** Mean, variance, and MSE prediction errors in the testing set. Boldface indicates best performance.

Predictor	SSCNNP	CNNP [28]	BIP [39]	MEDP [40]	GAP [41]
Mean	2.05	2.72	4.19	5.24	6.59
Variance	28.64	37.15	62.48	103.19	136.42
MSE	36.91	59.37	93.60	148.43	207.32

**Table 2 sensors-23-05360-t002:** Comparison of PSNR values (dB) for the watermarking capacities of 10,000 bits and 20,000 bits on the wind-turbine dataset. Boldface indicates best performance.

Image	10,000 bits					20,000 bits				
	BLTM [43]	IPPVO [25]	LPVO [42]	CNNP [28]	Proposed	BLTM [43]	IPPVO [25]	LPVO [42]	CNNP [28]	Proposed
Image (a)	60.17	59.93	60.29	60.42	60.68	55.76	55.97	55.66	55.94	56.02
Image (b)	58.27	58.72	58.26	58.53	58.62	55.24	55.23	55.33	55.39	55.48
Image (c)	58.51	58.46	58.51	58.58	58.67	55.38	55.31	55.40	55.42	55.47
Image (d)	58.59	58.56	58.66	59.05	59.14	55.10	55.13	55.12	55.27	55.47
Average (dB)	58.88	58.91	58.93	59.14	59.27	55.37	55.41	55.37	55.50	55.61

**Table 3 sensors-23-05360-t003:** Comparison of PSNR values (dB) for the watermarking capacities of 5000 bits and 10,000 bits on the substation-power-equipment dataset. Boldface indicates best performance.

Image	5000 bits					10,000 bits				
	BLTM [43]	IPPVO [25]	LPVO [42]	CNNP [28]	Proposed	BLTM [43]	IPPVO [25]	LPVO [42]	CNNP [28]	Proposed
image (a)	62.15	62.10	62.19	62.69	62.93	59.05	59.04	59.08	59.47	59.73
image (b)	56.39	56.27	56.60	57.04	57.26	53.83	53.94	54.36	54.84	55.09
image (c)	56.79	56.93	57.01	57.33	57.58	53.47	53.35	53.60	53.92	54.18
image (d)	62.03	62.04	62.27	62.38	62.54	58.63	59.07	59.14	59.31	59.44
Average(dB)	59.34	59.34	59.52	59.86	60.08	56.24	56.35	56.55	56.89	57.11

## Data Availability

Not applicable.

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
