# Peer review of "A Novel Steganography Method for Infrared Image Based on Smooth Wavelet Transform and Convolutional Neural Network"

_sensors, 2023, doi:10.3390/s23125360_

Round 1

Reviewer 1 Report

Attached

Good

Author Response

Hello. We have revised the thesis based on your proposed revisions. Attached are the responses to the revisions and the revised paper. Please take a look at them.

Reviewer 2 Report

·        Line 13, when an abbreviation is first used, it should be defined first. For instance, the acronym SSCNNP means What?

·        The authors should rewrite the abstract to follow this structure: background, objective, materials and methods, results, conclusion, and recommendations.

·        The authors should discuss the dataset used for the implementation. What are the attributes of the dataset, how was the dataset split, into what ratio, etc?

·        The authors should state how the study performance was accessed or evaluated.

·        How did the authors tune the optimal hyperparameter of all models? It should be described clearly.

·        How did you solve the problem of overfitting and small dataset

·        Overall, the English language and presentation style should be improved significantly. There were a lot of grammatical errors and typos. I suggest you have a colleague proficient in English and familiar with the subject matter review your manuscript or contact a professional editing service.

·        The limitation of the study should be stated, and they should present future research work.

·        The study should be compared with existing systems (state-of-the-art).

·        Source codes should be provided for replicating the study.

·        We are in May 2023, and I could see only 3 2022 citations and no 2023 citations. Citing recent literature have several advantages for both authors and journal. It can assist authors in establishing their credibility, demonstrating the relevance of their research, and help to avoid plagiarism. In the same way assists journals in increasing their visibility, improving their reputation, increasing their citation rates, and meeting reader expectations. Hence, for this reason, I have suggested some recent literature from the year 2023 relating to the study that you are to cite and reference in your article.

1.       Ao, J., Shao, X., Liu, Z., Liu, Q., Xia, J., Shi, Y.,... Ji, M. (2023). Stimulated Raman Scattering Microscopy Enables Gleason Scoring of Prostate Core Needle Biopsy by a Convolutional Neural Network. Cancer Research, 83(4), 641-651. doi: 10.1158/0008-5472.CAN-22-2146

2.       Fu, C., Yuan, H., Xu, H., Zhang, H., & Shen, L. (2023). TMSO-Net: Texture adaptive multi-scale observation for light field image depth estimation. Journal of Visual Communication and Image Representation, 90, 103731. doi: https://doi.org/10.1016/j.jvcir.2022.103731

3.       Wang, H., Cui, Z., Liu, R., Fang, L., & Sha, Y. (2023). A Multi-type Transferable Method for Missing Link Prediction in Heterogeneous Social Networks. IEEE Transactions on Knowledge and Data Engineering. doi: 10.1109/TKDE.2022.3233481

4.       Zhang, X., Huang, D., Li, H., Zhang, Y., Xia, Y.,... Liu, J. (2023). Self-training maximum classifier discrepancy for EEG emotion recognition. CAAI Transactions on Intelligence Technology. doi: https://doi.org/10.1049/cit2.12174

5.       Wang, S., Hu, X., Sun, J., & Liu, J. (2023). Hyperspectral anomaly detection using ensemble and robust collaborative representation. Information Sciences, 624, 748-760. doi: https://doi.org/10.1016/j.ins.2022.12.096

6.       Deng, Y., Zhang, W., Xu, W., Shen, Y., & Lam, W. (2023). Nonfactoid Question Answering as Query-Focused Summarization With Graph-Enhanced Multihop Inference. IEEE Transactions on Neural Networks and Learning Systems. doi: 10.1109/TNNLS.2023.3258413

7.       Liu, H., Xu, Y., & Chen, F. (2023). Sketch2Photo: Synthesizing photo-realistic images from sketches via global contexts. Engineering Applications of Artificial Intelligence, 117, 105608. doi: https://doi.org/10.1016/j.engappai.2022.105608

8.       Luo, H., Lou, Y., He, K., & Jiang, Z. (2023). Coupling in-situ synchrotron X-ray radiography and FT-IR spectroscopy reveal thermally-induced subsurface microstructure evolution of solid propellants. Combustion and Flame, 249, 112609. doi: https://doi.org/10.1016/j.combustflame.2022.112609

The English language and presentation style should be improved significantly. There were a lot of grammatical errors and typos. I suggest you have a colleague proficient in English and familiar with the subject matter review your manuscript or contact a professional editing service.

Author Response

Hello. We have read your comments carefully. The thesis has now been revised. The response to the comments and the revised paper are attached. Please take a look at them.

Round 2

Reviewer 2 Report

Thank you for attending to all my comments.

Minor editing of English language required

Author Response

After checking the English grammar and vocabulary of the full text, we have added data-based experimental results to the abstract. The articulation between the Introduction paragraphs has been optimized. New equation have been added, and some of the original equations have been presented in more detail.